# The Effect of Massage Force on Relieving Nonspecific Low Back Pain: A Randomized Controlled Trial

**DOI:** 10.3390/ijerph192013191

**Published:** 2022-10-13

**Authors:** Pei-Chun Chen, Li Wei, Chung-Yu Huang, Feng-Hang Chang, Yen-Nung Lin

**Affiliations:** 1Department of Physical Medicine and Rehabilitation, Wan Fang Hospital, Taipei Medical University, Taipei 116, Taiwan; 2Graduate Institute of Injury Prevention and Control, Taipei Medical University, Taipei 110, Taiwan; 3Taipei Neuroscience Institute, Taipei Medical University, Taipei 235, Taiwan; 4Division of Neurosurgery, Department of Surgery, Wan Fang Hospital, Taipei Medical University, Taipei 116, Taiwan; 5Department of Traditional Medicine, Wan Fang Hospital, Taipei Medical University, Taipei 116, Taiwan

**Keywords:** mechanical low back pain, musculoskeletal manipulation, myofascial pain syndrome, biomechanical phenomena, manual therapy

## Abstract

Objective: To investigate the effect of force applied during massage on relieving nonspecific low back pain (LBP). Methods: This single-blinded, randomized controlled trial enrolled 56 female patients with nonspecific LBP at a single medical center. For each participant, the therapist performed a 30 min massage session (20 min general massage and 10 min focal massage) using a special instrument with a force sensor inserted, for a total of six sessions in 3 weeks. During the 10 min focal massage, HF and LF groups received high force (HF, ≥2 kg) and low force (LF, ≤1 kg) massage, respectively. The primary outcome was pain intensity (i.e., visual analog scale (VAS), 0–10), and secondary outcomes comprised pain pressure threshold, trunk mobility, LBP-associated disability, and quality of life. Results: No significant between-group differences were observed in baseline characteristics. The HF group exhibited significantly lower VAS than did the LF group, with a mean difference of −1.33 points (95% CI: −2.17 to −0.5) at the end of the intervention, but no significant difference was noted at the end of the follow-up. A significant time effect (*p* < 0.05) was detected in all secondary outcomes except the pain pressure threshold and trunk mobility. A significant time × group interaction (*p* < 0.05) was found only for the VAS and pain pressure threshold. Conclusions: Compared with LF massage, HF massage exerted superior effects on pain relief in female patients with nonspecific LBP at the end of intervention. Applying different levels of force showed no effects on LBP-associated disabilities and quality of life.

## 1. Introduction

According to the findings of the Global Burden of Diseases, Injuries, and Risk Factors Study 2019, low back pain (LBP) remains the most prevalent health problem worldwide, contributing to the most years lived with disability [1]. Nonspecific LBP accounts for most cases of LBP, affecting females more than males [2]. Because nonspecific LBP does not have a known pathoanatomical cause, treatments are diverse and focus on symptom relief [2].

Massage is one of the therapeutic interventions that is most widely adopted around the globe for treating various musculoskeletal problems, including nonspecific LBP [3]. The practice of massage can be dated back thousands of years to ancient civilizations, including those in Babylonia, China, India, Greece, Egypt, and Rome [4]. Despite the long history and popularity of massage, limited efforts have been devoted to examining its effectiveness and underlying mechanisms. Several trials have investigated the effectiveness of specific massage styles on LBP, such as Thai [5,6], Swedish [7,8], Chinese massage [9], deep cross-friction massage [10], and myofascial release [11]. A Cochrane Review reported that the effects of massage for pain relief and functional improvement in individuals with subacute and chronic low back pain were significantly superior to those of inactive control interventions and were maintained at short-term follow-up but not long-term follow-up [12]. In a recent umbrella review of a diversity of interventions in treating low back pain, the effects of massage on pain and functions were considered inconclusive [13].

In the clinical setting, most massage styles involve direct pressure or friction on the soft tissue surface [14,15,16]. A practitioner applies mechanical force to transmit mechanical energy to the deep soft tissues. The amount of force applied is usually based on the practitioner’s preference and experience. It is unknown whether the efficacy of clinical massage depends on the force applied. No study has focused on the effects of applied force on the massage efficacy in treating LBP, possibly because of technical difficulties. Examining the effect of applied force can help improve the quality of clinical practice and facilitate the understanding of the biomechanical mechanisms of massage.

The aim of this study was to investigate the effect of massage force on treatment efficacy in female patients with nonspecific LBP by comparing two different force levels. We employed the “buffalo horn massage”, a style of Oriental massage using a rod-like instrument, and equipped the instrument with a force sensor for monitoring the applied force during massage. We hypothesized that higher force would generate more favorable outcomes (e.g., pain reduction and functional improvement) than lower force.

## 2. Materials and Methods

We recruited patients who visited the outpatient rehabilitation department of Wan Fang Hospital in Taipei between August 2018 and June 2020. Individuals who met the inclusion criteria and provided consent were randomly assigned to either the high-force (HF) group or the low-force (LF) group. A researcher applied randomization in blocks of 4 and conducted allocation by using concealed envelopes. All participants received massage therapy twice a week for a total of 6 sessions. Outcomes were evaluated before the intervention (week 0), after the intervention (week 4), and at follow-up (week 8). The outcome assessor was blinded to the treatment assignment. The study was approved by the Institutional Review Board of Taipei Medical University (reference number: N201709004) before the experiment was started, and has been conducted in accordance with the principles set forth in the Helsinki Declaration.

### 2.1. Participants

Women aged 20 to 65 years with diagnosed nonspecific LBP for more than 1 month were recruited. Individuals were excluded if they (1) were pregnant, (2) had LBP associated with systemic or specific disease (e.g., autoimmune, infectious, vascular, endocrine, metabolic, or neoplastic disease), (3) had received lumbar surgery, (4) presented with neurological signs (e.g., motor weakness and radiating pain), (5) had a skin condition on the back region (e.g., psoriasis, urticaria, or wounds), or (6) were receiving oral nonsteroidal anti-inflammatory drugs or steroid treatment for LBP at time of recruitment. The diagnosis of nonspecific LBP was based on history taking, physical examination, and spinal X-ray examinations. All participants were screened by a rehabilitation specialist and underwent physical and spinal X-ray examinations to exclude LBP with specific pathology (e.g., spondylolisthesis, fracture, severe spondylosis). Slightly degenerative changes of spine were acceptable.

### 2.2. Instrument

Figure 1 presents the massage instrument used in the “buffalo horn massage”, a rod-like instrument made from buffalo horn with 2 ends that differ in shape and contact area. The large end consisted of 5 rounded tips arranged in a circle. Each tip was 4 mm in diameter (Figure 1A). The small end was a single ball-tipped rod (ball diameter 8 mm; Figure 1B). A force sensor (JLBM-30 kg, Delta Transducers, USA) was inserted into the middle of the instrument to measure the vertical force transmitted through the instrument (Figure 1C). The therapist held one end of the instrument and contacted the participant’s skin with the other end, applying the force perpendicular to the skin. The value of the force applied was shown on the monitor (Figure 1D). The instrument had been validated by using a standard table weighing scale (MX-518, Honder Weighing Scale, Taiwan) to test the application of force ranging from 100 to 9000 g (90 sets) applied separately through each end of the massage instrument. The intraclass correlation coefficients for the large and small ends were 0.936 and 0.996, respectively.

### 2.3. Intervention

All participants received “buffalo horn massage” procedure with sessions consisting of 2 parts: a 20 min general massage with light strokes covering the entire area of the upper, middle, and lower back, followed by a 10 min focal massage targeting the region of the lower back identified by the participants as their main source of LBP. The treatment session was provided by a therapist with over 6 years of experience with “buffalo horn massage”. The intervention was performed in the outpatient rehabilitation department of Wan Fang Hospital.

In the first 20 min of the massage, both the HF and LF groups received the same protocol: the large end of the instrument (Figure 1A) was used to contact the skin, and the applied force (not exceeding 1 kg) was maintained as the instrument was moved down from the seventh cervical vertebra to the interior gluteal fold in a zigzag fashion. The maneuver was repeated until 20 min had elapsed. In the following 10 min, the massage was performed over the reported painful region using the small end of the rod (Figure 1B). The perpendicular force was applied to compress and rub the underlying soft tissue. During this part of the focal massage, the applied force was ≥2 kg in HF group and ≤1 kg in LF group. The therapist controlled the force using the feedback from the monitor (Figure 1D). The participants were placed in a prone position with their head nested in the hole of the massage table. Accordingly, they were unaware of the intervention process and the treatment assignment.

### 2.4. Outcome Measurements

The primary outcome was the pain intensity at the end of the intervention. The general pain intensity over the past 24 h was determined using the visual analog scale for pain (0–10). A pain reduction of 1.8 on the pain visual analog scale corresponds to a minimal clinically important difference in patients with chronic low back pain [17]. The secondary outcomes comprised the pain pressure threshold, trunk mobility, Oswestry Disability Index [18], Roland–Morris Low Back Pain Disability Questionnaire, [19] and the World Health Organization Quality of Life instrument (WHOQOL-BREF) [20].

The pain pressure threshold (i.e., the minimum amount of pressure that triggers pain) was quantified by applying the rubber probe tip (surface area 1 cm^2^) of a digital algometer (Force Ten FDX Force Gage, Wagner Instruments, USA) to the most painful region of the lower back [21]. The trunk mobility was assessed with modified Schober’s technique, by measuring the distance between 2 specific points along the spine when the participant reached maximal trunk flexion [22]. These clinical measurements were performed three times, and the average values were used for the analysis. The Oswestry Disability Index (scores range from 0 to 100%) [18] and Roland–Morris Questionnaire (0 to 24) [19] assess disability caused by low back pain, with higher scores indicating more severe disability. WHOQOL-BREF (4 to 20) [20] which covers 4 domains (physical, psychological, social relationships, and environment) assesses the general health-related quality of life.

At baseline (week 0) and postintervention (week 4), all outcomes were investigated by a researcher who was blinded to treatment allocation. At follow-up (week 8), outcomes were determined through the administration of a structured questionnaire (distributed by mail) comprising the pain VAS, ODI, RMQ, and WHOQOL-BREF.

### 2.5. Statistical Analysis

A review published in 2015 reported a pooled effect size of 0.75 between massage therapy and an inactive control intervention in achieving short-term pain reduction [12]. Based on this finding, we determined that the inclusion of 29 participants in each group could provide 80% power to detect between-group differences in the primary outcome at the postintervention assessment at a significance level of 0.05.

The chi-square test and independent *t* test were performed to compare the between-group distributions of the categorical and continuous variables, respectively. All continuous variables were tested using the Kolmogorov–Smirnov statistic, and results revealed normal distributions. For the primary outcome, analysis of covariance was performed to assess the between-group difference at the postintervention assessment with the adjustment of baseline values. The proportion of participants in each group whose pain reduction achieved the MCID on the pain VAS was also compared by chi-square test. For the secondary outcomes, repeated measures of analysis of variance were conducted to assess time and group interactions. The significant time × group interaction was subjected to a post hoc test. An intention-to-treat analysis was primarily conducted, and missing data were imputed using the last observation carried forward method. A per-protocol analysis was also performed. A 2-tailed *p* value of <0.05 was considered statistically significant.

To investigate potential responders to massage treatment, subgroup analyses were conducted on body mass index (BMI), body fat percentage, pain duration, age, PPT, ODI, and RMQ. The cutoff value was 23 for BMI, as a value over 23 indicated overweight for Asian people [23], and 20% for ODI, which distinguished mild and moderate (or higher) disability [24]. For variables lacking references for cutoffs (e.g., age, body fat percentage, PPT, and RMQ), the medians of these variables were set as the cutoff values. A between-group comparison of pain VAS scores at the postintervention assessment was conducted for each subgroup. The effect size was calculated to assess the between-group difference, with absolute values of 0.2 to 0.5, 0.5 to 0.8, and >0.8 representing small, moderate, and large effect sizes, respectively [13,25]. Statistical analysis was performed using SPSS version 18.0 (SPSS Inc., Chicago, IL, USA).

### 2.6. Data Availability

The data associated with the paper are not publicly available but are available from the corresponding author on reasonable request.

## 3. Results

In total, 51 (91.1%) of the 56 patients completed the intervention and postintervention assessments, and 50 (89.3%) patients completed the follow-up (Figure 2). The average age of the participants was 42.3 years, and the mean pain duration was 9.6 months. The mean scores on the pain VAS, ODI, and RMQ were 4.6, 19.5 points, and 7.4 points, respectively. No significant differences in baseline characteristics were observed between the groups (Table 1). No significant differences in baseline characteristics were observed between the participants who completed the study (n = 50) and who did not (n = 6).

Table 2 presents the results on pain, disability, and quality of life. For the primary outcome, a significant between-group difference of −1.33 (95% CI: −2.17 to −0.5; *p* = 0.002) in pain VAS scores was noted at the end of the intervention. A significantly higher number of participants in the HF group had a pain VAS reduction reaching MCID than in the LF group (i.e., 17 vs. 9; *p* = 0.032). For the secondary outcomes, a significant time effect was observed in all measurements except for PPT and trunk mobility. A significant time × group interaction was found only for the pain VAS and PPT. The post hoc tests revealed a significant between-group difference of 1.00 kg/cm^2^ in PPT at the postintervention assessment. The between-group difference on the pain VAS was nonsignificant at the end of follow-up. The results from per-protocol analysis were very similar (Appendix A).

The results of subgroup analyses on pain VAS scores in week 4 are shown as forest plots (Figure 3). Participants who were younger (i.e., ≤40 y), had a lower BMI (≤23 kg/m^2^), had a lower body fat percentage (≤30%), had a lower PPT (≤5.1 kg/cm^2^), and had milder pain-related disability (ODI ≤20% and RMQ ≤6 points) exhibited a significant between-group difference favoring HF massage while their counter subgroups did not.

## 4. Discussion

To our knowledge, this is the first study that aims to investigate the effects of force applied in massage on treating LBP. We compared the effects of HF (i.e., ≥2 kg) and LF (i.e., ≤1 kg) applied during a 10 min focal massage included in six 30 min sessions. Massage with HF generated greater pain reduction than with LF after intervention. However, the effect did not last up to the follow-up. Massage force was not found to be associated with the LBP-associated disabilities and quality of life. Participants who were younger, had a lower BMI or body fat percentage, had a lower PPT, and had milder LBP-associated disability were more likely to benefit from HF massage in terms of pain relief at the end of intervention. These findings contribute to current knowledge about the massage applied in treating LBP.

The association between the massage force and the effect on pain relief has not been explored, possibly due to the lack of proper tools to measure the applied force during massage. We were able to measure the applied force by inserting a force sensor in the massage instrument. Because of the lack of research for reference, the definition of HF and LF in the present study was empirical. Prior to the investigation, we noted that the force applied during the first 20 min light strokes of each session was generally less than 1 kg. We also observed that the applied force level in HF group during the focal massage usually ranged from 2 to 6 kg and rarely reached 8 kg. Thus, this specific benchmark (i.e., ≥2 kg and ≤1 kg) was adopted to define the HF and LF applied during the focal massage and was expected to produce a discrepancy.

Given the definition of HF and LF, the HF group showed greater pain reduction than the LF group, as evidenced by the significantly lower VAS pain value and higher number of participants who reached the MCID for VAS pain at the end of intervention. These findings were accompanied by significantly greater elevation of pain threshold in the HF group at week 4. The between-group difference in PPT was 1.0 kg/cm^2^ at week 4, which exceeded the reported minimal detectable change for PPT for spinal disorders [26]. These findings indicate a short-term effect on pain relief attributable to applied force during massage, in line with previous studies that showed the effect of massage on pain reduction was short-term [7,13].

There were no significant differences between HF and LF groups regarding the majority of secondary outcomes. Several facts might be considered. The overall Oswestry Disability Index was 19.5% at baseline, indicating that a population with mild low back pain-associated disabilities (i.e., Oswestry Disability Index < 20%) [24] had been enrolled. The floor effect may have occurred. In addition, the painful sites of LBP subjects were focally treated for only 10 min within each 30 min session. The duration and frequency of the treatment might have been insufficient to produce discernible effects. Furthermore, the LF group may have received effective massage treatment, thus serving as a strong active control and potentially reducing the between-group difference in LBP-related disabilities and quality of life.

Previous studies addressing the effect of massage on pain relief [7,11,12,27,28,29,30] have not explored the potential responders. Therefore, we conducted the subgroup analyses in a hypothesis-generating manner. Notably, participants who had a lesser BMI, lesser body fat, lower PPT, and milder pain-related disability tended to respond to HF massage. Overweight subjects with a higher body fat percentage could have had a thicker layer of adipose tissue over the muscles. Massage in these participants would require the application of a higher level of force to allow greater depth of mechanical force transmission to the target soft tissue. In other words, a greater level of force (than that used in our HF massage) may be needed to generate desirable effects in this population. The reason why participants with lower PPT responded to HF massage is also unclear. Subjects with lower PPT might perceive greater discomfort during HF massage. Therefore, it could be possible that pain relief was associated with desensitization by repeated local stimulation in the painful area through massage maneuver [31,32,33]. Apparently, further research is needed to unveil the mechanism of pain relief by massage.

## 5. Limitations

There are some limitations in this study. First, we used a special instrument and massage technique. There was a gap in extrapolation from our results to other styles of massage. The efficacy of applied force may be affected by the used instruments or body parts (e.g., finger, knuckle, elbow techniques, etc.), considering the different shape and contact area between different massages [34], and variations in patient physique. Second, we only enrolled female patients for the homogeneity of the sample, which could limit the generalization of our results if there are intersexual differences. Third, the PPT and trunk mobility were not measured at 8 weeks. Finally, the psychological effect of massage, which is considered a crucial factor in pain reduction, was not investigated in this study.

## 6. Conclusions

Compared with LF massage, HF massage exerted superior effects on pain relief in female patients with nonspecific LBP at the end of intervention, but the effect did not last to the 8-week follow-up time point. The force applied during massage exerted no effects on LBP-associated disabilities and quality of life.

## Figures and Tables

**Figure 1 ijerph-19-13191-f001:**
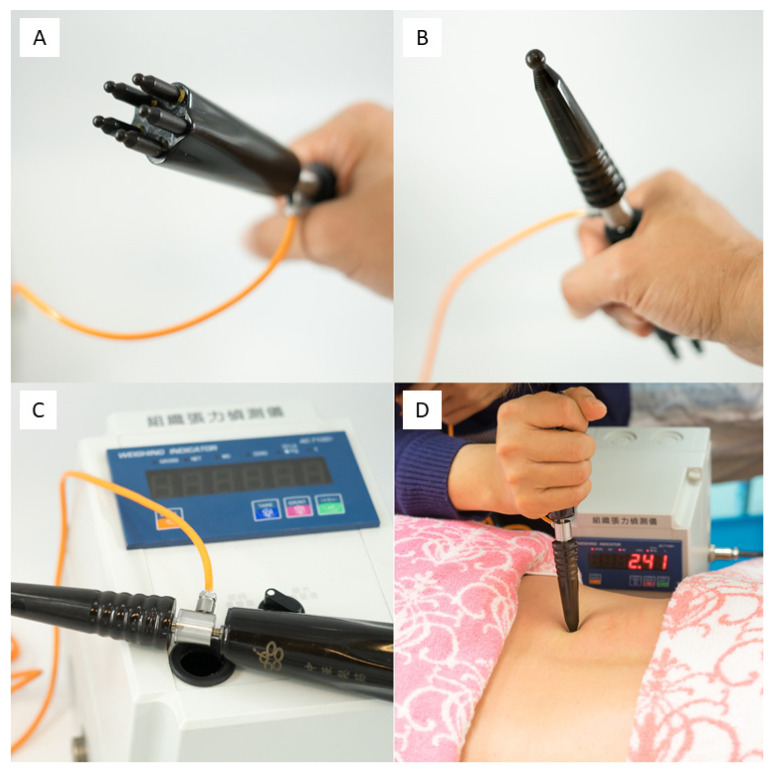
Photographs of the Massage Instrument. (**A**) The large end. (**B**) The small end. (**C**) The force sensor inserted into the middle of the instrument. (**D**) The therapist applied force perpendicularly to participants’ backs, and the monitor displayed the value of the applied force.

**Figure 2 ijerph-19-13191-f002:**
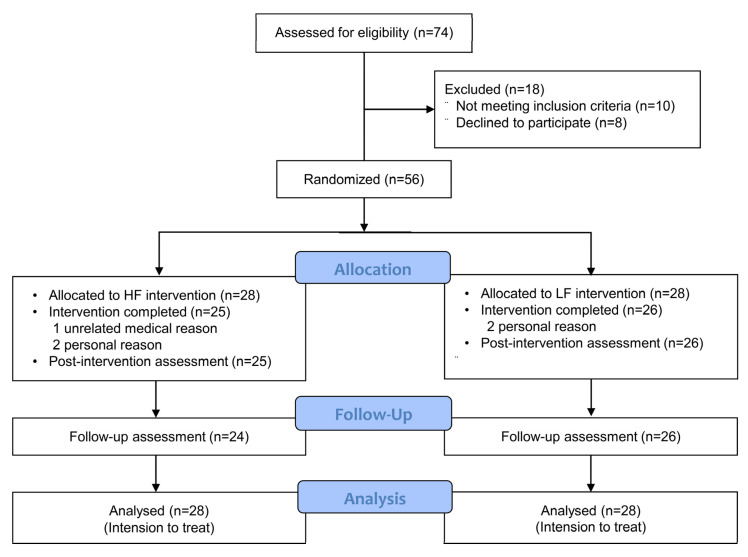
Study Flowchart.

**Figure 3 ijerph-19-13191-f003:**
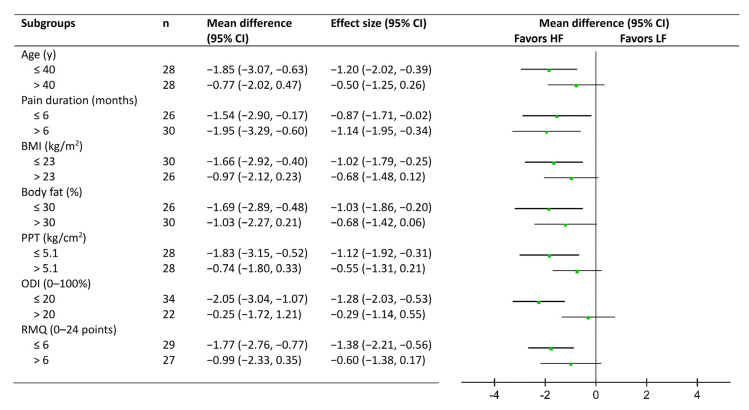
Subgroup Analysis of the Effect of Applied Force on Short-Term Pain Reduction.

**Table 1 ijerph-19-13191-t001:** Baseline Characteristics of the Participants.

Variables	HF Group (n = 28)	LF Group (n = 28)	*p* Value
Age	42.4 ± 13.07	42.1 ± 13.32	0.936
Pain duration (mo)	10.4 ± 7.1	8.8 ± 9.4	0.482
BMI (kg/m^2^)	23.90 ± 4.39	22.71 ± 3.18	0.252
Body fat (%)	29.55 ± 8.09	29.89 ± 3.89	0.840
Pain VAS (0–10)	4.36 ± 1.61	4.76 ± 1.35	0.318
PPT (kg/cm^2^)	5.52 ± 2.07	5.48 ± 2.26	0.955
ODI (0–100%)	18.69 ± 12.90	20.36 ± 13.37	0.637
RMQ (0–24)	7.39 ± 4.95	7.39 ± 4.52	1.000
Trunk anterior flexibility (cm)	20.06 ± 1.30	20.40 ±1.01	0.280
WHOQOL-BREF (4–20)			
Physical	13.35 ± 1.88	13.00 ± 2.45	0.554
Total score	13.43 ± 1.72	13.07 ± 1.50	0.411
Education level			0.301
High school or lower	2	6	
College/university	22	18	
Graduate school	4	4	
Occupation			0.721
None	5 (17.9%)	3 (10.7%)	
Laborious	12 (42.9%)	14 (50%)	
Nonlaborious	11 (39.3%)	11 (39.3%)	

BMI, body mass index; HF, high-force; LF, low-force; PPT, pain pressure threshold; ODI, Oswestry Disability Index; RMQ, Roland–Morris Low Back Pain Disability Questionnaire; pain VAS, visual analog scale for pain; WHOQOL-BREF, short form of the World Health Organization Quality of Life instrument. Values are presented as means ± standard deviations or as frequencies (percentages).

**Table 2 ijerph-19-13191-t002:** Effects of Massage Force on the Study Outcomes.

Variables	HF Group n = 28	LF Group n = 28	Mean Difference (95% CI)	*p* Value ^b^	Time *p* Value ^c^	Time × Group *p* Value ^c^
Primary outcome						
Pain VAS score in wk 4	2.29 ± 1.51	3.79 ± 1.89	−1.33 (−2.17 to −0.5)	0.002		
VAS-MCID in wk 4 ^a^	17 (60.7%)	9 (32.1%)		0.032		
Secondary outcome						
Pain VAS score					<0.001	0.021
Baseline	4.36 ± 1.61	4.76 ± 1.35				
Change in wk 4	−2.15 ± 1.54	−0.97 ± 1.79	−1.18 (−2.07 to −0.28)			
Change in wk 8	−1.52 ± 1.75	−1.39 ± 1.74	−0.14 (−1.07 to 0.80)			
PPT (kg/cm^2^) ^d^					0.108	0.042
Baseline	5.52 ± 2.07	5.48 ± 2.26				
Change in wk 4	0.89 ± 1.72	−0.11 ± 1.87	1.00 (0.04 to 1.97)			
Trunk mobility (cm) ^d^					0.960	0.992
Baseline	20.06 ± 1.30	20.40 ±1.01				
Change in wk 4	0.01 ± 1.64	0.01 ± 1.01				
ODI					<0.001	0.858
Baseline	18.69 ± 12.90	20.36 ± 13.37				
Change in wk 4	−4.11 ± 5.81	−2.54 ± 6.33				
Change in wk 8	−3.39 ± 6.67	−2.75 ± 4.99				
RMQ					0.001	0.417
Baseline	7.39 ± 4.95	7.39 ± 4.52				
Change in wk 4	−1.71 ± 2.93	−1.61 ± 5.65				
Change in wk 8	−1.39 ± 3.21	−2.04 ± 4.96				
WHOQOL-physical					<0.001	0.509
Baseline	13.35 ± 1.88	13.00 ±2.45				
Change in wk 4	1.23 ± 2.12	0.78 ± 1.70				
Change in wk 8	0.85 ± 1.85	0.81 ± 1.61				
WHOQOL-total					<0.001	0.195
Baseline	13.43 ±1.72	13.07 ±1.50				
Change in wk 4	1.15 ± 1.23	0.67 ± 1.27				
Change in wk 8	0.77 ± 1.11	0.25 ± 1.07				

HF, high-force; LF, low-force; PPT, pain pressure threshold; ODI, Oswestry Disability Index; RMQ, Roland–Morris Low Back Pain Disability Questionnaire; pain VAS, visual analog scale for pain; WHOQOL-BREF, the short version of the World Health Organization Quality of Life questionnaire. Values are presented as the mean ± standard deviation or frequency (percentage). ^a^ VAS-MCID, the number of participants whose pain reduction reached the minimal clinically important difference on the VAS pain. ^b^ Between-group comparison of the primary outcome, either by ANCOVA or chi-square tests. ^c^ By repeated measures ANOVA. ^d^ PPT and trunk mobility were not evaluated in week 8.

## Data Availability

The data presented in this study are available on request from the corresponding author. Public data sharing is not applicable to this article due to ethical considerations and privacy restrictions.

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
