# Peer review of "The Effect of Massage Force on Relieving Nonspecific Low Back Pain: A Randomized Controlled Trial"

_ijerph, 2022, doi:10.3390/ijerph192013191_

Round 1

Reviewer 1 Report

1. How do you drive the exclusion criteria for this study? 

2. What is the reliability of the instruments (force sensors, weighing scale, etc) used in the study?

3. Add a figure which illustrates the massage's parts properly 

4. In the 2.3 section, Why did you choose 20-min general? Why not 22-mints? The duration of the massage is a very crucial factor. Justification of the duration which has been selected for the intervention. Selection of random massage duration can't be considered for scientific study. 

5. Quality of Figure 3 has to be improved 

6. Justify the sample consider for this study. 

Author Response

  1. How do you drive the exclusion criteria for this study? 

Response: Thanks for the comment. I supposed you are asking how we checked if the participant met the exclusion criteria. All the participants were arranged a meeting by a physiatrist of the rehabilitation department of Wan-Fang hospital to check the medical history, X-ray, and do physical examinations. Please refer to Line 80-82.

  1. What is the reliability of the instruments (force sensors, weighing scale, etc) used in the study?

Response: Thanks for the comment. We understand the importance of the reliability of a studied instrument (ie, the massage instrument) in scientific research. To test it, we used a standard table weighing scale as an anchor. The intraclass correlation coefficients between the massage instrument and the standard table weighing scale was reported (Line 92 to Line 96). This is what we can do for the reliability for the studied instrument.

  1. Add a figure which illustrates the massage's parts properly

Response: I suppose you are requesting a figure that illustrates the body part where the massage is performed. However, the location of the massage has been clearly stated in Line 101-Line 104. The requested figure may not be necessary.

  1. In the 2.3 section, Why did you choose 20-min general? Why not 22-mints? The duration of the massage is a very crucial factor. Justification of the duration which has been selected for the intervention. Selection of random massage duration can't be considered for scientific study. 

Response: Thanks for the comment. The duration of the massage is determined empirically, which follows the protocol of “Baffalo horn massage”. We agree that the duration of massage is crucial. But we doubt the designing of massage protocol of any existing research is evidence-based.

  1. Quality of Figure 3 has to be improved 

Response: Thank you for the comment. We slightly revised the figure.

  1. Justify the sample consider for this study. 

Response: I suppose the comment is about the determination of study population. We recruited the participants who sustained non-specific low back pain for more than 1 month. The concern regarding the study population may be that we recruited only female patients. The decision is made to decrease the heterogeneity of treatment response between gender and to increase the statistical power. However, such decision limits the generalization of our results if there are intersexual differences. We have addressed such limitation in Line 278-Line 279.

Reviewer 2 Report

The Authors present their original and interesting study on the effects of the force applied in the massage in the treatment of patients with non specific low back pain (LBP). The effect of massage on LBP is yet to be defined, and this study is a commendable effort to elucidate certain aspects of massage effects, and possibly help to better understand the underlying mechanisms. This is a randomized trial in with two groups of LBP patients were treated with different massage's forces, by means a force-measuring "buffalo horn massage" device. The conception of the study is smart, and the methods is sound. The results are interesting, though not outstanding, and consistent with the study design and method. The results of the study are well discussed, with acknowledgement of some limitations.

However, the Authors should better explain the inclusion criteria, namely the definition of nonspecific LBP, when Spinal X-ray is concerned as an imaging exam to rule out spinal pathology. In fact, with a mean age of 42 +/-13, is very likely to find even if minimal degenerative changes on spine X-ray. Then, the Authors should explain whether inclusion criteria included an absolutely normal X-ray, or was some degree of degenerative changes accepted.

Author Response

The Authors should better explain the inclusion criteria, namely the definition of nonspecific LBP, when Spinal X-ray is concerned as an imaging exam to rule out spinal pathology. In fact, with a mean age of 42 +/-13, is very likely to find even if minimal degenerative changes on spine X-ray. Then, the Authors should explain whether inclusion criteria included an absolutely normal X-ray, or was some degree of degenerative changes accepted.

Response: Thanks for the comment. Now we added some descriptions in the Participants part to make it clearer. (Line 80-Line 85)

Round 2

Reviewer 1 Report

I appreciate the effort you put into the revised manuscript. However, I'm not able to understand clearly the determination of the sample size. 

Thank you